



# An EARLINET Early Warning System for atmospheric aerosol aviation hazards

Nikolaos Papagiannopoulos[1,2], Giuseppe D'Amico[1], Anna Gialitaki[3,4], Nicolae Ajtai[5], Lucas Alados-Arboledas[6], Aldo Amodeo[1], Vassilis Amiridis[3], Holger Baars[7], Dimitris Balis[4], Ioannis Binietoglou[8], Adolfo Comerón[2], Davide Dionisi[9], Alfredo Falconieri[1], Patrick Freville[10], Anna Kampouri[3,4], Inna Mattis[11], Zoran Mijić[12], Francisco Molero[13], Alex Papayannis[14], Gelsomina Pappalardo[1], Alejandro Rodríguez-Gómez[2], Stavros Solomos[3], and Lucia Mona[1]

[1]Consiglio Nazionale delle Ricerche - Istituto di Metodologie per l'Analisi Ambientale (CNR-IMAA), C.da S. Loja, Tito Scalo (PZ), Italy
[2]CommSensLab, Dept. of Signal Theory and Communications, Universitat Politècnica de Catalunya, Barcelona, Spain
[3]IAASARS, National Observatory of Athens, Athens, Greece
[4]Laboratory of Atmospheric Physics, Physics Department, Aristotle University of Thessaloniki, Thessaloniki, Greece
[5]Babes-Bolyai University of Cluj Napoca, Cluj, Romania
[6]Department of Applied Physics, University of Granada, Granada, Spain
[7]Leibniz Institute for Tropospheric Research (TROPOS), Leipzig, Germany
[8]National Institute of R&D for Optoelectronics (INOE), Magurele, Romania
[9]Consiglio Nazionale delle Ricerche - Istituto di Scienze Marine (CNR-ISMAR), Roma, Italy
[10]Observatoire de Physique du Globle (OPGC-LaMP), Clermont-Ferrand, France
[11]Deutscher Wetterdienst, Meteorologisches Observatorium Hohenpeißenberg, Germany
[12]Institute of Physics Belgrade, University of Belgrade, Belgrade, Serbia
[13]Centro de Investigaciones Energéticas, Medioambientales y Tecnológicas, Department of Environment, Madrid, Spain
[14]Laser Remote Sensing Unit, Physics Dept., National Technical University of Athens, Athens, Greece

**Correspondence:** Nikos Papagiannopoulos (nikolaos.papagiannopoulos@imaa.cnr.it)

**Abstract.** A stand-alone lidar-based method for detecting airborne hazards for aviation in near-real-time (NRT) is presented. A polarization lidar allows for the identification of irregular-shaped particles such as volcanic dust and desert dust. The Single Calculus Chain (SCC) of the European Aerosol Lidar Network (EARLINET) delivers high resolution pre-processed data: the calibrated total attenuated backscatter and the calibrated volume linear depolarization ratio time series. From these calibrated

5    lidar signals, the particle backscatter coefficient and the particle depolarization ratio can be derived in temporally- high resolution, and thus provide the basis of the NRT Early Warning System (EWS). In particular, an iterative method for the retrieval of the particle backscatter is implemented. This improved capability was designed as a pilot that will produce alerts for imminent threats for aviation. The method is applied to data during two diverse aerosol scenarios: first, a record breaking desert dust intrusion in March 2018 over Finokalia, Greece, and, second, an intrusion of volcanic particles originating from Mount Etna in

10    June 2019 over Antikythera, Greece. Additionally, a devoted observational period including several EARLINET lidar systems demonstrates the network's preparedness to offer insight into natural hazards that affect the aviation sector.



# 1   Introduction

During the aviation crisis related to the volcanic eruption of Eyjafjallajökull in 2010, the European Aerosol Research Lidar Network (EARLINET; Pappalardo et al., 2014) provided range-resolved information to the World Meteorological Organization (WMO) on a daily basis (reports available at: www.earlinet.org, last access: 31 October 2019). The reports communicated the

altitude, time, and location of the volcanic clouds over Europe. Furthermore, the time- height evolution of the lidar returns was freely available in near- real-time (NRT) on the EARLINET website. The non-automated, non-harmonized, and non-homogenized process and the lack of tailored products for natural hazards made the EARLINET data disregarded in the decision making process.

The lessons learned from the Eyjafjallajökull crisis emphasized the vulnerability of air transportation to natural hazards

(Bolic and Sivcev, 2011). Volcanic ash plumes as well as desert dust outbreaks present an imminent threat to aviation as they lead, among others, to poor visibility with considerable consequences to flight operations (Bolic and Sivcev, 2011; Middleton, 2017). Aircraft that do fly in volcanic/desert dust conditions can have a variety of damages from scouring of surfaces to engine failure (Eliasson et al., 2016). The aftermath of an encounter can be immediate, reducing flight safety, furthermore it can financially affect the airlines due to higher maintenance costs and replacement of mechanical equipment.

Furthermore, the Eyjafjallajökull eruption highlighted the gap in the availability of real-time measurements and monitoring information for airborne hazards. Specifically, it became evident the lack of height-resolved information, a key aspect in flight planning and mitigation strategies. In the frame of the H-2020 research project EUNADICS-AV (European Natural Disaster and Information System for Aviation; www.eunadics.eu, last access: 31 October 2019) funded by the European Commission, different organizations worked together in a consortium to provide relevant data during situations when aviation is affected by

airborne hazards (i.e., volcanic ash, desert dust, biomass burning, radionuclide). Crucial for the overall success of the project and the EWS design were the review of the available observations and the collection of specific requirements from the different stakeholders that once more pointed out the importance of height-resolved information.

A polarization lidar is an important tool to characterize the different aerosols. This system permits the discrimination of light- depolarizing coarse-mode particles such as volcanic and desert dust and fine-mode particles such as smoke particles

and anthropogenic pollution (e.g., Tesche et al., 2011; Mamouri and Ansmann, 2017). Further, the lidar set-up allows for the retrieval of coarse-mode and fine-mode backscatter coefficient for wavelengths 532 nm and 1064 nm (e.g., Tesche et al., 2009). When synergistically used with a photometer, it is possible to retrieve their mass concentration profile (e.g., Ansmann et al., 2012; Lopatin et al., 2013; Chaikovsky et al., 2016).

During the last years, EARLINET has increased strongly its observing capacity with the addition of new stations and system

upgrade, namely, the installation of depolarization channels. Besides, the further development of the Single Calculus Chain (D'Amico et al., 2015, 2016; Mattis et al., 2016) under the ACTRIS (Aerosol, Clouds and Trace gases Research InfraStructure Network) umbrella eliminated the inconsistencies in the retrieval procedures and in the signal error calculation, and automated the data evaluation, and now allows for the NRT data processing and the generation of tailored products network-wide. EAR-LINET has already demonstrated the networks' NRT capabilities as well as assisted modelling studies in NRT evaluation and





assimilation (Wang et al., 2014; Sicard et al., 2015). As a consequence, EARLINET is prepared to provide promptly height-resolved information and tailored products that were highly missed during the 2010 aviation crisis. Therefore, a methodology for an Early Warning System (EWS) based solely on EARLINET data is developed.

In Sect. 2 we present the EARLINET remote sensing network and the data that we used in this study. In Sect. 3 we introduce the methodology of the EARLINET-based EWS. In Sect. 4 we present the results obtained by applying the methodology to real measurements and the lessons learned from a multi- station EARLINET observational period. Finally, in Sect. 5 we give our conclusions and indicate directions for future work.

## 2 EARLINET

The European Aerosol Research Lidar Network (EARLINET; Pappalardo et al., 2014) was established in 2000, providing aerosol profiling data on a continental scale, and is now part of the Aerosols, Clouds, and Trace gases Research InfraStructure (ACTRIS; www.actris.eu, last access: 31 October 2019). Nowadays, more than 30 stations are active and perform measurements according to the network's schedule (one daytime and two night-time measurements per week). Further measurements are devoted to special events, such as volcanic eruptions, forest fires, and desert dust outbreaks (e.g., Mona et al., 2012; Pappalardo et al., 2013; Ortiz-Amezcua et al., 2017; Granados-Muñoz et al., 2016). The majority of the EARLINET stations operate multi-wavelength Raman lidars, that combine a set of elastic and nitrogen inelastic channels and are equipped with depolarization channels. This lidar configuration allows for the retrieval of intensive aerosol profiles, such as particle lidar ratio, particle Ångström exponent, and particle depolarization ratio. These variables are shown to vary with the aerosol type and location and, consequently, EARLINET stations are able to characterize the aerosol load (Müller et al., 2007). Accordingly, EARLINET has established tools for the automatic aerosol characterization (Nicolae et al., 2018; Papagiannopoulos et al., 2018).

To ensure homogeneous, traceable, and quality controlled analysis of raw lidar data across the network, a centralized and fully automated analysis tool, called the Single Calculus Chain (SCC), has been developed within EARLINET. Raw lidar data are first submitted to the central SCC server by each EARLINET station and several lidar products are generated automatically. In particuar, low resolution (in both time and space) uncalibrated pre-processed products provided by the SCC EARLINET Lidar Pre-processor (ELPP) module (D'Amico et al., 2016) and aerosol optical properties vertical profiles provided by the SCC EARLINET Lidar data Analyzer (ELDA) module (Mattis et al., 2016) are made available. Recently a new version of the SCC has been released providing also standardized high-resolution pre-processed lidar products. These new products include the calibrated attenuated backscatter coefficient, and volume linear depolarization ratio time series at instrumental time and space resolution. Particular attention has been paid to the calibration of the high resolution products: an automatic and fully traceable calibration procedure using the low resolution SCC-retrieved particle backscatter and extinction coefficients has been designed and implemented in the SCC framework.

The cloud screening module is responsible for the cloud identification in uncalibrated lidar signals, especially low clouds, since such clouds do not permit the aerosol optical property retrieval by ELDA. Note that the cloud removal is also essential





in our EWS methodology. The input of the algorithm are the high resolution pre-processed signals produced by the SCC HiRELPP (High Resolution EARLINET Pre- Processor) module. The current cloud screening detects clouds as bins with irregularly high values in signal and edge strength (Nixon and Aguado, 2019; Tramutoli, 1998). The algorithm works well with uncalibrated signal recorded by multiple lidar systems across EARLINET. However, false detection of aerosol-laden bins as

cloud can occur, especially in cases where there is high contrast between an aerosol layer and the rest of the atmosphere. For this reason, the development of a cloud screening module based on calibrated lidar signals and quantitative criteria is foreseen.

The calibrated high resolution data along with the cloud screening output are essential for the proposed methodology and are used in the EWS.

## 2.1    The site of Finokalia and Antikythera, Greece

The EARLINET component of NOA (National Observatory of Athens) for the period of April 2017 until May 2018 had deployed the NOA lidar system on the north coast of Crete. The Finokalia Atmospheric observatory (35.34 N, 25.67 E) is a research infrastructure with activities covering in situ aerosol characterization, 3–D aerosol distribution, and gas precursors. Since June 2018, the system is relocated in the island of Antikythera, where a suite of remote sensing sensors are installed in order to study the properties of natural aerosol particles (i.e., sea salt, dust, volcanic ash) in Mediterranean background

conditions. The islands of Crete and Antikythera are very often affected by windblown dust originating from the Sahara, due to their proximity to the African coastline and can be across the travelled path of volcanic dust and sulfate aerosols from the Italian active volcanoes (e.g., Hughes et al., 2016).

The NOA lidar system Polly$^{XT}$ (e.g., Engelmann et al., 2016) operates in the frame of EARLINET, and under the umbrella of ACTRIS. The system is equipped with three elastic channels at 355 nm, 532 nm and 1064 nm, two vibration-rotation Raman

channels at 387 nm and 607 nm, two linear depolarization channels at 355 nm and 532 nm, and one water vapour channel at 407 nm. Depending on the atmospheric conditions, the combined use of its near range and far range telescopes provides reliable vertical profiles of aerosol optical properties from 0.2–0.4 km to almost 16 km in height.

## 2.2    Additional data

For the detection of the desert dust plume, satellite imagery from the Spinning Enhanced Visible Infra-Red Imager (SEVIRI) is

used. SEVIRI is a line-by-line scanning radiometer onboard the Meteosat Second Generation (MSG) geostationary satellite. It provides data in 12 spectral bands every 15 min for the full Earth disc area. The spatial resolution is around 3 km at the nadir, apart from the High- Resolution Visible (HRV) band (1 km). In this study, we used a largely accepted multi-temporal scheme of satellite data analysis (Tramutoli, 2007) to detect the dust plume over the Mediterranean basin. In particular, we used the eRST$_{DUST}$ (enhanced Robust Satellite Technique for Dust Detection) algorithm (Marchese et al., 2017), which combines an

index analysing the visible radiance (at around 0.6 μm) to another one based on the brightness temperature difference (BTD) of the signal measured in the SEVIRI spectral channels centred at 10.8 μm and 12 μm wavelength.

For the detection of the volcanic dust we use the Lagrangian transport model FLEXPART (FLEXible PARTicle dispersion model; Brioude et al., 2013; Stohl et al., 2005) in a forward mode to simulate the dispersion of volcanic emissions from Etna.





Dispersion simulations are driven by hourly meteorological fields from the Weather Research and Forecasting model (WRF; Skamarock et al., 2008) at $36 \times 36$ km horizontal resolution. The initial and boundary conditions for the off-line coupled WRF-FLEXPART runs are taken from the National Center for Environmental Prediction (NCEP) final analysis (FNL) dataset at a $1° \times 1°$ resolution at 6-hourly intervals. The sea surface temperature (SST) is taken from the NCEP $0.5° \times 0.5°$ analysis. The simulated case study did not include an eruptive stage; therefore the initial injection height is set from the crater level (3.3 km a.s.l.) up to 4 km a.s.l. A total of 10000 tracer particles are released for this simulation. Dry and wet deposition processes are also enabled in these runs. Saharan dust transport is also described in WRF with the Air Force Weather Agency (AFWA) scheme (Jones et al., 2012).

## 3 Methodology

### 3.1 Retrieval of the particle parameters in temporally high resolution

The delivery of an alert using EARLINET data is based on a two-step approach. In the first step, the high resolution calibrated data are used to estimate the particle backscatter coefficient and the particle linear depolarization ratio. In order to retrieve the particle backscatter coefficient, an iterative methodology is adapted. The methodology, described in Di Girolamo et al. (1999), is able to retrieve particle backscatter coefficient with an overall error of no more than 50%. Prior to that, the cloud contaminated pixels are removed from the data using the cloud screening algorithm developed for the SCC (see Section 2).

The method is similar to that of Mattis et al. (2016) that SCC employs to derive optical products from elastic backscatter signals. For an ever-available NRT and automated aerosol retrieval we use channels for elastic backscattering, including depolarization, since Raman observations during daytime are hitherto challenging.

The calibrated attenuated backscatter coefficient provided by the SCC can be written as

$$\beta_{att}(\lambda,r) = [\beta_{mol}(\lambda,r) + \beta_{par}(\lambda,r)]\, T_{mol}^2(\lambda,r)T_{par}^2(\lambda,r) \tag{1}$$

where $\beta_{par}(\lambda,r)$ and $\beta_{mol}(\lambda,r)$ are, respectively, the backscatter coefficient for particles (par) and molecules (mol); and $T_{par}^2(\lambda,r)$ and $T_{mol}^2(\lambda,r)$ represent the two-way attenuation to and from range $r$ due to, respectively, particles and molecules at wavelength $\lambda$. The latter and can be expressed as

$$T_{par/mol}^2(\lambda,r) = exp\left[-2\int_0^R \alpha_{par/mol}(\lambda,r)dr\right] \tag{2}$$

where $\alpha_{par}(\lambda,r)$ and $\alpha_{mol}(\lambda,r)$ are the particle and molecular extinction coefficient, respectively. The term $\lambda$ is omitted from the subsequent expressions as the analysis explicitly focuses on 532 nm. The terms $\alpha_{mol}(r)$ and $\beta_{mol}(r)$ can be estimated from temperature and pressure profiles.





In an initial step, the attenuation in the atmosphere is neglected, $\alpha_{par}^{(0)}(r)=0\,\mathrm{m}^{-1} \Rightarrow T_{par}^{(0)^2}(r)=1$, which reduces Eq. 1 to

$$\beta_{par}^{(1)}(r) = \beta_{mol}(r)\left[\frac{\beta_{att}(r)}{\beta_{mol}(r)T_{mol}^2(r)} - 1\right] \tag{3}$$

The particle extinction coefficient is estimated by multiplying $\beta_{par}^{(1)}(r)$ with a constant lidar ratio, $S_{par}$. Using the particle extinction coefficient in Eq. 1 we derive a new backscatter coefficient given by

$$\beta_{par}^{(2)}(r) = \beta_{mol}(r)\left[\frac{\beta_{att}(r)}{\beta_{mol}(r)T_{mol}^2(r)T_{par}^{(1)^2}(r)} - 1\right] \tag{4}$$

Baars et al. (2017) developed a method to derive of atmospheric parameters in temporally high resolution and refer to the product of Eq. 4 as the quasi-particle backscatter coefficient, which serves as best estimate for the particle backscatter coefficient. However, here, the particle backscatter

$$\beta_{par}^{(i)}(r) = \beta_{mol}(r)\left[\frac{\beta_{att}(r)}{\beta_{mol}(r)T_{mol}^2(r)T_{par}^{(i-1)^2}(r)} - 1\right] \tag{5}$$

is calculated in the i–th iteration step from the calibrated attenuated backscatter coefficient. The procedure is successfully terminated if the absolute difference between the backscatter coefficient of two subsequent profiles is smaller than a fixed threshold. The absolute difference, $\Delta_\beta$, is defined as

$$\Delta\beta^{(i)} = \left|\int \beta_{par}^{(i)}dr - \int \beta_{par}^{(i-1)}dr\right| \tag{6}$$

We found that less than 10 steps are required for a difference of 1 % for the cases examined herein.

The particle depolarization ratio at 532 nm can be defined as (Baars et al., 2017)

$$\delta_{par} = [\delta_{vol}(r)+1]\times\left(\frac{\beta_{mol}(r)\left[\delta_{mol}-\delta_{vol}(r)\right]}{\beta_{par}(r)\left[1+\delta_{mol}\right]}\right)^{-1} - 1 \tag{7}$$

where $\delta_{mol}$ is the molecular depolarization ratio and is calculated theoretically (Behrendt and Nakamura, 2002). The term $\delta_{vol}(r)$ denotes the volume depolarization ratio and it is the output of SCC.

The input lidar ratio value used in the retrieval could significantly affect the results. Papagiannopoulos et al. (2018) used 48 ± 13 sr for fresh volcanic particles and 55 ± 7 sr for desert dust particles observed over EARLINET sites in their aerosol classification, which illustrates the variability of this intensive parameter. The uncertainty induced due to the assumption of lidar ratio can easily exceed 20 % (Sasano et al., 1985) and presents an important source that affects the retrieval. In this study, $S_{par}$=50 sr is chosen for the backscatter coefficient retrieval, as it is a good compromise for many EARLINET sites and different aerosol conditions (Papayannis et al., 2008; Müller et al., 2007; Mona et al., 2014; Papagiannopoulos et al., 2016).





Figure 1 shows a desert dust layer around $3\,\mathrm{km}$ over the Potenza EARLINET station on 4 April 2016, 18:47–22:15 UTC. The backscatter coefficient at $532\,\mathrm{nm}$ retrieved for $30\,\mathrm{sr}$, $50\,\mathrm{sr}$, and $70\,\mathrm{sr}$ along with the backscatter coefficient from the Raman method is shown (Fig. 1a). The three curves almost coincide in the upper part (relative difference is around $5\,\%$) and deviate from one another by less than $35\,\%$ in the lower portion of the profile, where local aerosol is mixed with dust particles.

The performance of the iterative method for $S_{par}$=$50\,\mathrm{sr}$ can be assessed in Fig. 1b. The overall agreement is very good with the relative difference being around $4\,\%$, however the iterative method underestimates almost everywhere the Raman method due to the assumption of $S_{par}$=$50\,\mathrm{sr}$, instead of the measured $43 \pm 7\,\mathrm{sr}$. Figure 1c highlights the effect when the directly measured lidar ratio is plotted against the fixed lidar ratio. Evidently, the curves agree fairly well for the aerosol layer (i.e., desert dust) in the free troposphere and deviate from the layer below (i.e., values over $50\,\mathrm{sr}$). As discussed above, the inference

of the lidar ratio is an important factor, yet a lidar ratio value valid for a common volcanic dust and desert dust layer will provide a robust solution for this approach.

## 3.2   Aviation alert delivery

In the second step, the location and the intensity of the volcanic dust and desert dust event are identified. Mona and Marenco (2016) reported particle depolarization ratio values around $35\,\%$ for freshly emitted particles from various volcanoes and

that the values decrease with time. Similarly, pure Saharan dust particles are supposed to have a slightly smaller particle depolarization ratio of $31\,\%$ (Freudenthaler et al., 2009). Since nonspherical particles such as volcanic and desert dust particles yield high particle depolarization ratio values, the one-step polarization-lidar photometer networking (POLIPHON) method is used (e.g., Ansmann et al., 2012).

The particle depolarization ratio is used to separate the nonspherical particles contribution to the particle backscatter coeffi-

cient. Mamouri and Ansmann (2014) describe in detail the retrieval process, however, here, we treat volcanic dust and desert dust inextricably. The volcanic dust and desert dust backscatter coefficient can be expressed by

$$\beta_c = \beta_{par} \frac{(\delta_{par} - \delta_{nc})(1 + \delta_c)}{(\delta_c - \delta_{nc})(1 + \delta_{par})} \tag{8}$$

with the coarse ($c$) and non-coarse ($nc$) depolarization ratios set to $\delta_c = 0.31$ and $\delta_{nc} = 0.05$, respectively. For values $\delta_{par} < \delta_{nc}$, we need to set $\beta_{nc} = \beta_{par}$. Similarly, when $\delta_{par} > \delta_c$ we set $\beta_c = \beta_{par}$.

Until the aviation crisis in 2010 the planes were advised to avoid the volcanic plumes regardless of the aerosol concentration (Guffanti et al., 2010). Recently, the International Civil Aviation Organization (ICAO, 2014) has established three ash concentration thresholds which play a key role in the decision making process. Aircraft are allowed to fly below $0.2\,\mathrm{mg/m^3}$, whereas over $2\,\mathrm{mg/m^3}$ and $4\,\mathrm{mg/m^3}$ (depending on the aircraft's resilience) they are forbidden to fly.

The methodology proposed by Ansmann et al. (2012) for the estimation of aerosol mass concentration profiles employs

data from a single-wavelength polarization lidar. The methodology retrieves mass concentration profiles with an uncertainty $20$–$30\,\%$ and has proven to be robust and applicable to very different scenarios (e.g., Mamali et al., 2018; Córdoba-Jabonero et al., 2018) that needs one wavelength and can be applied to cloudy skies. We chose to convert the three ash concentration





thresholds into particle backscatter coefficient. The threshold values for the particle backscatter coefficient ($\beta_{th}$) are estimated as

$$\beta_{th} = M \frac{1}{\rho c_v S} \tag{9}$$

where $M$ is the mass concentration given by ICAO, $\rho$ the volcanic and desert dust bulk density, $c_v$ the mass-to-extinction

conversion factor, and $S$ the volcanic and desert dust lidar ratio. All the terms have to be assumed constant and they are selected from literature. The above concentration thresholds (e.g., 0.2, 2, 4 mg/m³) are used for the term $M$. For the $\rho$, we used the value $2.6 \, \mathrm{g/cm^3}$ that corresponds to a commonly used value for volcanic and desert dust applications (e.g., Gasteiger et al., 2011; Ansmann et al., 2012; Binietoglou et al., 2015; Mamali et al., 2018). The term $S$ is chosen to be $50 \, \mathrm{sr}$ as a good compromise for fresh volcanic particles (e.g., Ansmann et al., 2011) and Saharan dust (e.g., Wiegner et al., 2012).

The term $c_v$ can be estimated using AERONET observations, although for an EWS and day-night availability we have to select a constant value for volcanic dust and desert dust. Figure 2 shows an overview of AERONET-based $c_v$ values. To interpret the horizontal axis of the figure, one should also look at Tabl. 1. The figure is separated in volcanic (grey points) and desert dust (orange points) and depicts the range of the observed values, furthermore the plot shows the mean and standard deviation for the all-average of the conversion factors. It is evident from Fig. 2 that for both volcanic and desert dust the values accumulate

between $0.6$–$0.9 \times 10^{-6} \, \mathrm{m}$ with a mean of $(0.76 \pm 0.06) \times 10^{-6} \, \mathrm{m}$. It is worth noting that although most of the conversion factors were estimated using carefully selected AERONET observations, Mamouri and Ansmann (2017) and Ansmann et al. (2019) use a climatology to derive the conversion factor.

The conversion factor for the coarse particles (i.e., volcanic and desert dust) varies strongly with the distance from the source and, in the case of volcanic eruptions, with the eruption type. Ansmann et al. (2012) highlight that when particles larger

than $15 \, \mu\mathrm{m}$ (i.e., the higher limit of the assumed particles radii for the AERONET data analysis scheme) are present the mass concentration may be underestimated by more than a 100 %. The conversion factor in case of dense and coarser plumes should be much higher, and, consequently will have adverse impact in our EWS approach. For instance, Pisani et al. (2012) used a conversion factor of $0.6 \times 10^{-5} \, \mathrm{m}$ for fresh erupted volcanic plume near the Mount Etna in Italy. A similar increase, although less pronounced, in the conversion factor can be observed in Mamouri and Ansmann (2017) and Ansmann et al. (2019), where

the authors retrieve a dust coarse-mode conversion factor (i.e., the values reported in Fig. 2). It is believed that particles bigger than $10 \, \mu\mathrm{m}$ usually fall quickly to the ground, whereas smaller particles can travel over long distances (Goudie and Middleton, 2006; Wilson et al., 2012). Conversely, van der Does et al. (2016) and Ryder et al. (2018) have illustrated that the desert dust size far away from its source is much coarser than previously suggested and incorporated into climate models. In light of the above, we chose as the conversion factor in our approach the maximum retrieved value, which is $0.9 \times 10^{-6} \, \mathrm{m}$ (Ansmann

et al., 2012). Hence, the thresholds for the particle backscatter coefficient become $1.7 \times 10^{-6} \, \mathrm{m^{-1} sr^{-1}}$ (for $0.2 \, \mathrm{mg/m^3}$), $1.7 \times 10^{-5} \, \mathrm{m^{-1} sr^{-1}}$ (for $2 \, \mathrm{mg/m^3}$) and $3.4 \times 10^{-5} \, \mathrm{m^{-1} sr^{-1}}$ (for $4 \, \mathrm{mg/m^3}$). Given also that the EARLINET stations are far from the active European volcanoes (i.e., Etna and the Icelandic volcanoes), we consider that the selected AERONET-derived conversion factor holds for most of the situations.





Figure 3 illustrates the decision flowchart for the aviation alert delivery where three alert levels are available: Low alert ($0.2 < M_c < 2\,\mathrm{mg/m^3}$), Medium level alert ($2 < M_c < 4\,\mathrm{mg/m^3}$), and High level alert ($M_c > 4\,\mathrm{mg/m^3}$) indicating the increasing amount of dust particles likely dangerous for flight operations. The coarse backscatter coefficient due to the highly depolarizing particles is estimated first. Next, the coarse backscatter coefficient is checked and the level of alert is decided. A similar

methodology has been demonstrated within an international demonstration exercise for the purpose of the EUNADICS-AV project, where an artificial Etna eruption was simulated (Hirtl et al., 2019).

## 4 Results

In this section we apply the described methodology to potential perilous events recently detected by the stations of Finokalia and Antikythera, Greece. The observations refer to the same lidar system that was initially deployed in Finokalia and later migrated

to the island of Antikythera. The aim is not to present a detailed analysis of investigated cases, but instead to demonstrate the potential of this methodology to be integrated as a tailored EARLINET product for the fast alerting of airborne hazards relevant to flight operations.

### 4.1 Desert dust particles case

During March 2018, frequent intense dust storms affected Greece with the region of Libya being the originating source

(Kaskaoutis et al., 2019). Strong surface, and mid- and upper-troposphere Khamsin winds transported dust northwards for 4 distinct periods (i.e., 4–7, 17, 21–22, 25–26 March). Solomos et al. (2018) examined in detail the record-breaking episode of 21–22 March, where surface concentrations exceeded $6\,\mathrm{mg/m^3}$ on 22 March and resulted in the closure of the Heraklion airport.

Here, we focus on the 21 March when the dust cloud initially appeared over Crete. Figure 4 shows the dust map derived from

20 SEVIRI data along with the cloud cover at 12:00 UTC. The dusty pixels are depicted in two different colours as a function of the confidence levels of the dust detection scheme (i.e., brown means high confidence and orange mid-low confidence). In particular, the dust cloud moves from North Africa towards the Eastern Mediterranean, where the cloud cover impedes the dust detection over the insular Greece, although the map demonstrates the intensity and the geographic extent of the dust event.

The coarse particle backscatter coefficient, the particle depolarization ratio at $532\,\mathrm{nm}$, as described in Sect. 3.1, the cloud

mask, and the tailored product for the period 07:00–13:00 UTC are shown in Fig. 5. The dust particles arrive over Finokalia around 08:00 UTC in a filament-like layer about $4\,\mathrm{km}$, wherein the dust particles exhibit high values of the particle depolarization ratio. Figure 5d shows the alert product for aviation, which demonstrates 'Low level' alert indicating considerable amount of dust particles in the troposphere likely dangerous for flight operations. In particular, the coarse particle backscatter coefficient at $532\,\mathrm{nm}$ exhibits values up to $6 \times 10^{-6}\,\mathrm{m^{-1}sr^{-1}}$, which exceeds the threshold value of $1.7 \times 10^{-6}\,\mathrm{m^{-1}sr^{-1}}$.

Besides, this case illustrates the advantage of a ground-based lidar system to operate below high clouds that obstruct satellite observations (see Fig. 4) and, therefore, provide important insight.





As the event aggravated in the following hours, the lidar signal is most likely attenuated highlighting the limitation of the methodology. However, the alert delivery could act as a pre-alerting tool for aviation pinpointing the specific aerosol conditions. A similar approach for airport operations has been developed using automatic lidars and ceilometers for the prediction of fog formation (Haeffelin et al., 2016).

## 4.2 Volcanic and desert dust particles case

The eruption of volcano Mount Etna which began in the early hours of 30 May, 2019, injected ash in the atmosphere in the altitude of 3.5–4.0 km (VAAC Toulouse report at 11:21 UTC, 30 May). The volcanic activity ceased most likely on 3 June (https://ingvvulcani.wordpress.com, last access: 31 October 2019). This volcanic activity did not lead to any air traffic disruption, as was the case for the explosion on 20 July. The latter caused flights re-routing and delays (source: *La Repubblica*).

Aerosol particles of possibly volcanic origin were monitored with the multi-wavelength lidar of NOA over Antikythera, Greece. The presence of these elevated layers above Greece could be a result of the continuous Etna activity for the past few days. Figure 6 shows two distinct layers with different characteristics for the period from 21:00 UTC on the 2 June to 06:00 UTC on the 3 June. The first layer is first observed between 1–2 km on the 2 June and remains visible for the rest of the temporal window. The particle backscatter coefficient is around $1 \times 10^{-6}\,\mathrm{m}^{-1}\mathrm{sr}^{-1}$ and the particle depolarization ratio is below 5 % and differentiates from the second layer aloft. The second layer is seen after 23:30 UTC on 2 June until 03:00 UTC 3 June and resides in the range 2–3 km. The layer particle depolarization ratio is well above 20 % and indicates non-spherical particles. Moreover, it exhibits higher particle backscatter coefficient ($\sim 3 \times 10^{-6}\,\mathrm{m}^{-1}\mathrm{sr}^{-1}$). As a result, the alert is triggered for the latter. It is noteworthy that, as seen in the cloud mask, few pixels within the same aerosol layer are wrongly classified as clouds. The improvement of the cloud masking module is currently ongoing and is expected to eliminate false cloud detection, but nonetheless the aerosol layer is very well captured by the method.

The identification of the source of the two aerosol layers is made through an analysis of FLEXPART and WRF-Chem simulations. Figure 7 indicates the eastward transport of a relatively thin ($\sim$60 km horizontal width) volcanic ash plume from Mt. Etna towards Greece. The simulated plume is misplaced by about 70 km towards the north from the EARLINET Antikythera station, however, its vertical structureis still evident in the cross-section of Fig. 8. The eastward motion and the vertical profile of simulated aerosol volcanic plume corroborate the existence of volcanic particles in the upper layer of Fig. 6. The non-depolarizing structures below 2 km are sea-salt particles possibly mixed with dust particles. Limited concentrations ($>0.04\,\mathrm{mg/m^3}$) of dust are simulated at these heights by the WRF-Chem model (Fig. 9) accompanied by increased relative humidity near the surface, thus implying hygroscopic growth and more spherical particles in this area. In synthesis, both observations and model simulations advocate for the co-existence of volcanic dust and aged desert dust particles in the aerosol scene. Consequently, the alert delivered refers to volcanic dust.

## 4.3 Lessons learned from the EUNADICS-AV exercise

The application of the EWS and the timeliness delivery of the EARLINET data were tested in real-time during the EUNADICS-AV exercise, where EARLINET stations performed synchronous measurements. The EUNADICS-AV demonstration exercise





in March 2019 based on a fictitious volcanic eruption demonstrated that tailored observations as well as model services can profitably support aviation stakeholders (Hirtl et al., 2019).

In particular, 13 EARLINET stations contributed to the exercise according to a predefined measurements schedule – i.e., from 11:00 to 17:00 UTC on 5 March 2019 and from 07:00 to 11:00 UTC on 6 March 2019 – independently of the station's

capabilities with respect to the EWS. This decision stems from the opportunity to assess the sequence of procedures for the real-time data retrieval and data visualization. Besides the measurements schedule, the stations submitted raw lidar data in the SCC server every hour, which were automatically available on the EARLINET quicklook interface (https://quicklooks.earlinet.org/, last access: 16 January 2019). For the majority of the stations and temporal windows low clouds and cirrus clouds were observed. Table 2 summarizes the measurements gathered per hour segment and the station capabilities with respect to the

EWS. In total, 73% of the measurements were performed successfully, whereas rain and manning the stations mostly inhibited the rest. Moreover, only for 6 of the stations it was possible to retrieve the tailored product mainly because of the lack of the depolarization information during the exercise. The tailored product did not produce any alert as the aerosol layers were neither volcanic dust nor desert dust nor yielded high backscatter coefficient values.

Overall, the raw lidar data were streamed and processed in less than 30 minutes from the measurement enabling the timeli-

15 ness delivery of the lidar data and the tailored product, where possible. Furthermore, the demonstration exercise was the first occasion in which the proposed methodology was tested in NRT and the obtained results suggest that the network could support stakeholders actively in decision making during an aviation crisis.

## 5   Conclusions

A tailored product for aviation hazards by means of high resolution lidar data has been proposed for the first time to our

knowledge. Especially, the methodology employs single-wavelength EARLINET high resolution data (i.e., 532 nm calibrated backscatter coefficient and 532 nm calibrated volume linear depolarization ratio) and yields NRT alerts based on established aerosol mass concentration thresholds. The methodology aims to provide an EARLINET EWS for the fast alerting of airborne hazards exploiting the SCC advancements and to mitigate the effects of a future aviation crisis. The application on EARLINET data from eastern Mediterranean demonstrated the strength of the methodology to identify possibly dangerous for the aviation

volcanic ash and desert dust plumes.

One of the key challenges for a NRT automated alert delivery is the calibration of the backscatter and depolarization profiles as the elastic and depolarization channels are used. The EARLINET SCC ensures the absolute calibration of the lidar signals. As a source of high uncertainties in the retrieval of the particle backscatter coefficient, the inference of the lidar ratio was acknowledged. Accordingly, an iterative method has been developed to work with high-resolution lidar data and compares

well with particle backscatter coefficient profiles retrieved with the Raman method.

Additionally, and equally important in the alert delivery approach is the conversion factor with which the mass concentration thresholds are converted into particle backscatter coefficient. The AERONET-derived conversion factors are known to be restricted by the AERONET data inversion scheme and underestimate large to giant particles. Therefore, the selected conversion



factor was chosen (i.e., $0.9 \times 10^{-6}$ m) as the maximum value of the literature review with reference to fresh volcanic and desert dust observations.

The NRT operation of EARLINET during the EUNADICS-AV exercise was successfully demonstrated. successful application of the method in NRT has been achieved during the EUNADICS-AV exercise. The raw data upon uploaded to the SCC server were automatically processed and became freely accessible through the EARLINET portal and available in order to initiate the alert delivery. The exercise demonstrated the strength of the network, which if promptly triggered can enable measurements in case of natural hazards for aviation.

Besides, a similar approach can be extended to lidar systems operated by the European volcano observatories. Two examples of such observatories in Europe are the Istituto Nazionale di Geofisica e Vulcanologia – Osservatorio Etneo (INGV-OE) and the Icelandic Meteorological Office (IMO). INGV-OE is responsible for monitoring of Mt. Etna while IMO is responsible for monitoring all volcanic activity in Iceland.

This method is highly versatile as it can adapt to other wavelengths and the aerosol backscatter thresholds can be set to accommodate different volcanic and desert dust scenarios by adjusting the conversion factor, the lidar ratio, the bulk density, and the mass concentration levels. Besides, even if developed on the basis of EARLINET, it can be applied to lidar systems as those that are part of GALION (AD-Net, LALINET, MPLNET) as well as to current (CALIPSO; Cloud-Aerosol Lidar and Infrared Pathfinder Satellite Observations) and future lidar-based satellite missions (EarthCARE; Earth Cloud, Aerosol and Radiation Explorer).

*Data availability.* The data can be found in https://data.earlinet.org.

*Author contributions.* The conceptualisation and design of this study were carried out by NP and LM. GD is the lead scientist and curator of the EARLINET SCC data. IM and IB created the calibration and cloud mask module for the EARLINET SCC, respectively. VA and AG are the PI and data originator for the EARLINET stations of Finokalia and Antikythera, respectively. SS and AK performed FLEXPART model simulations for the Antikythera case study. AF retrieved the dust product from SEVIRI data for the Finokalia case study. AA, AC, AP, ARG, DD, DM, FM, HB, IM, LAA, NA, PF, VM, and ZM are either the PIs or the key personnel of the stations involved in the measurements exercise and ensured the high quality operation of the respective lidars. The interpretation of results was determined from discussions involving all authors. The original draft of the paper was written by NP, and reviewed and edited by all the co-authors.

*Competing interests.* The authors declare that they have no conflict of interest.

*Acknowledgements.* The authors acknowledge EARLINET for providing aerosol lidar profiles (www.earlinet.org, last access: 31 October 2019). We thank the ACTRIS-2 and ACTRIS Preparatory Phase projects that have received funding from the European Union's Horizon 2020





research and innovation programme (grant agreement No. 654109) and from European Union's Horizon 2020 Coordination and Support Action (grant agreement No. 739530), respectively. This work has been conducted within the framework of the EUNADICS-AV project, which has received funding from the European Union's Horizon 2020 research programme for Societal challenges - smart, green and integrated transport under grant agreement No. 723986. Furthermore, the research leading to these results has received funding from the COST Action

5    CA16202, supported by COST Association (European Cooperation in Science and Technology). e-shape (EuroGEOSS Showcases: Applications Powered by Europe), a project funded under the European Union's Horizon 2020 Programme (Grant Agreement n. 820852), aiming at the development and uptake of 27 cloud-based pilot applications, addressing the Sustainable Development Goals, The Paris Agreement and the Senda Framework, is also acknowledged. Part of the work performed for this study was funded by the Ministry of Research and Innovation through Program I - Development of the national research-development system, Subprogram 1.2 - Institutional Performance - Projects

10   of Excellence Financing in RDI, Contract No.19PFE/17.10.2018 and by Romanian National Core Program Contract No.18N/2019. VA acknowledges support of this work by the the European Research Council (ERC) under the European Community's Horizon 2020 research and innovation framework program – ERC grant agreement 725698 (D-10 TECT).



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



**Table 1.** The code used in Fig. 2 and the respective reference.

| Code | Reference |
| --- | --- |
| V1 | Ansmann et al. (2010) |
| V2 | Ansmann et al. (2011) |
| V3 | Ansmann et al. (2012) |
| V4 | Devenish et al. (2012) |
| V5 | Sicard et al. (2012) |
| D1 | Ansmann et al. (2012) |
| D2 | Binietoglou (2014) |
| D3 | Córdoba-Jabonero et al. (2018) |
| D4 | Mamali et al. (2018) |
| D5 | Mamouri and Ansmann (2014) |
| D6 | Mamouri and Ansmann (2017) |
| D7 | Ansmann et al. (2019) |





**Table 2.** EARLINET stations that participated in the EUNADICS-AV exercise during 5–6 March 2019. It is reported the percentage of the measurements made for the two consecutive days and the specific temporal windows. The 'x' denotes the stations for which it was possible to derive the alert for aviation - i.e., availability of calibrated backscatter coefficient and depolarization ratio at 532 nm. The ($^*$) indicates the stations equipped with depolarization channel although this information was not available during the exercise.

| EARLINET station | Measurements performed [%] | | EWS |
| --- | --- | --- | --- |
| | 05/03, 11–17 UTC | 06/03, 7–12 UTC | |
| Antikythera (GR) | 100 | 100 | x |
| Athen$s^*$ (GR) | 100 | 100 | |
| Barcelona (ES) | 100 | 0 | x |
| Belgrade (SRB) | 100 | 100 | |
| Clermont-Ferrand$^*$ (FR) | 33 | 40 | |
| Cluj$^*$ (RO) | 100 | 80 | |
| Granada (ES) | 17 | 20 | x |
| Hohenpeissenberg (DE) | 100 | 100 | x |
| Leipzig (DE) | 100 | 100 | x |
| Madrid (ES) | 33 | 0 | |
| Potenza (IT) | 100 | 100 | x |
| Roma - Tor Vergata (IT) | 100 | 100 | |
| Thessaloniki$^*$ (GR) | 83 | 100 | |



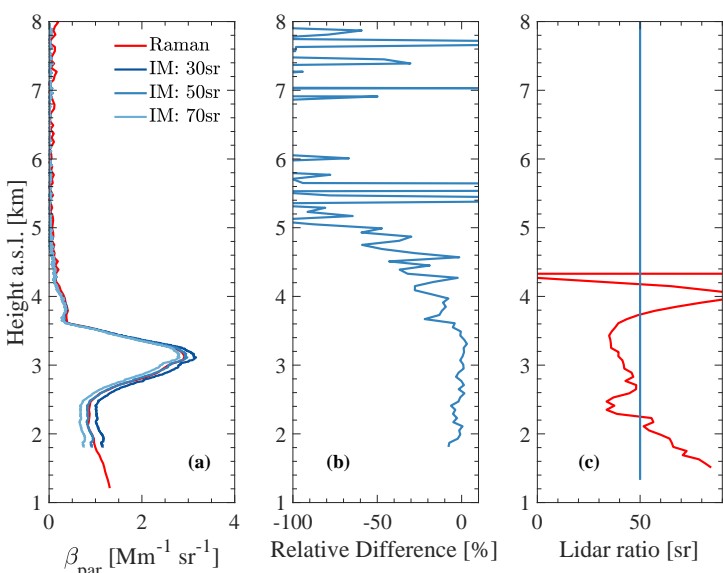

**Figure 1. (a)** The 532 nm backscatter coefficient retrieved with the iterative method (IM) for 30 sr, 50 sr, and 70 sr along with the backscatter coefficient determined with the Raman method (standard SCC product) measured at Potenza (760 m a.s.l.), Italy, on 4 April 2016, 18:47–22:15 UTC. The lidar system of Potenza has a full overlap at around 1.15 km a.s.l. for 532 nm (Madonna et al., 2018). **(b)** The relative difference between the iterative method (IM: 50 sr) and the Raman method backscatter coefficient. **(c)** The lidar ratio profile measured with the Raman method and the fixed lidar ratio used for the iterative method.



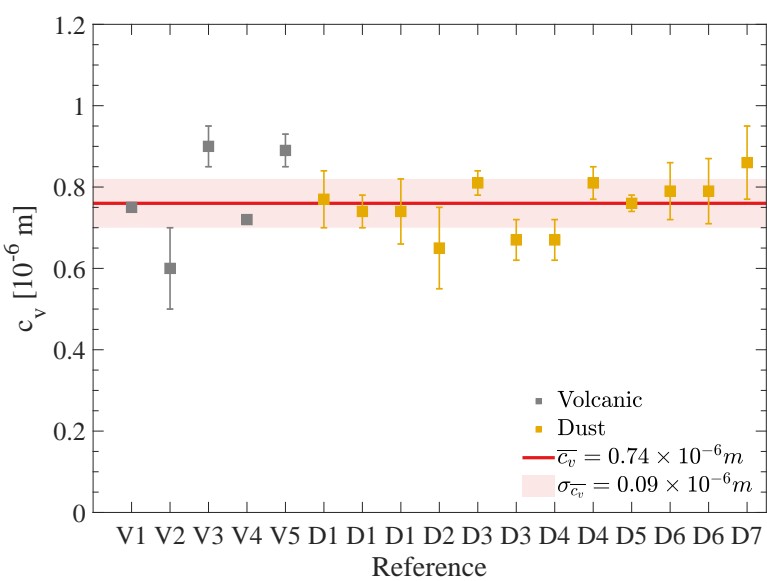

**Figure 2.** The scatter plot indicates the mean and the standard deviation of the conversion factor $c_v$ for the different literature references. The plot is color coded with respect to 'Volcanic' (grey) and 'Dust' (orange) observations. The red line highlights the all-mean conversion factor and the reddish-pink rectangle shows the standard deviation – i.e., $(0.76 \pm 0.06) \times 10^{-6}$ m.



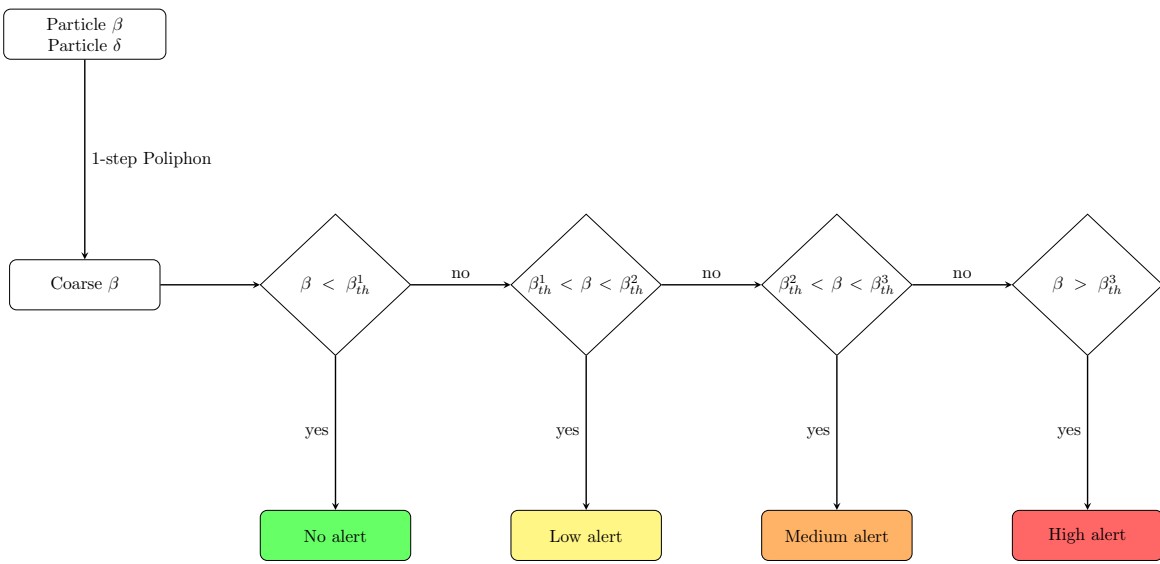

**Figure 3.** The EARLINET alert delivery scheme for aviation. The particle backscatter coefficient and depolarization ratio are used to estimate the coarse backscatter coefficient (one–step POLIPHON method). Three levels are considered that correspond to 'Low alert' for particle concentrations higher than $0.2\,\mathrm{mg/m^3}$ and lower than $2\,\mathrm{mg/m^3}$, 'Medium level alert' for concentration higher than $2\,\mathrm{mg/m^3}$ and lower than $4\,\mathrm{mg/m^3}$, and 'High level alert' for mass concentration higher than $4\,\mathrm{mg/m^3}$. The three backscatter coefficient thresholds are: $\beta_{th}^1 = 1.7 \times 10^{-6}\,\mathrm{m^{-1}sr^{-1}}$, $\beta_{th}^2 = 1.7 \times 10^{-5}\,\mathrm{m^{-1}sr^{-1}}$, and $\beta_{th}^3 = 3.4 \times 10^{-5}\,\mathrm{m^{-1}sr^{-1}}$.

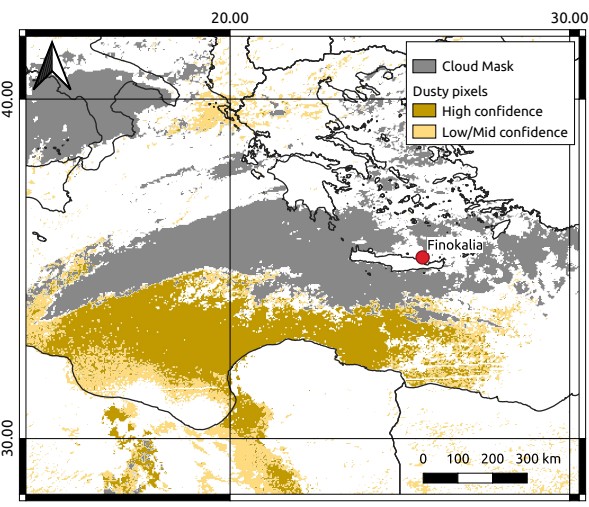

**Figure 4.** The dust SEVIRI product (Marchese et al., 2017) at 12:00 UTC on 21 March 2018 is represented in confidence levels (i.e., brown pixels refer to high confidence and orange pixels to mid-low confidence). Additionally, the gray pixels indicate the cloud cover.



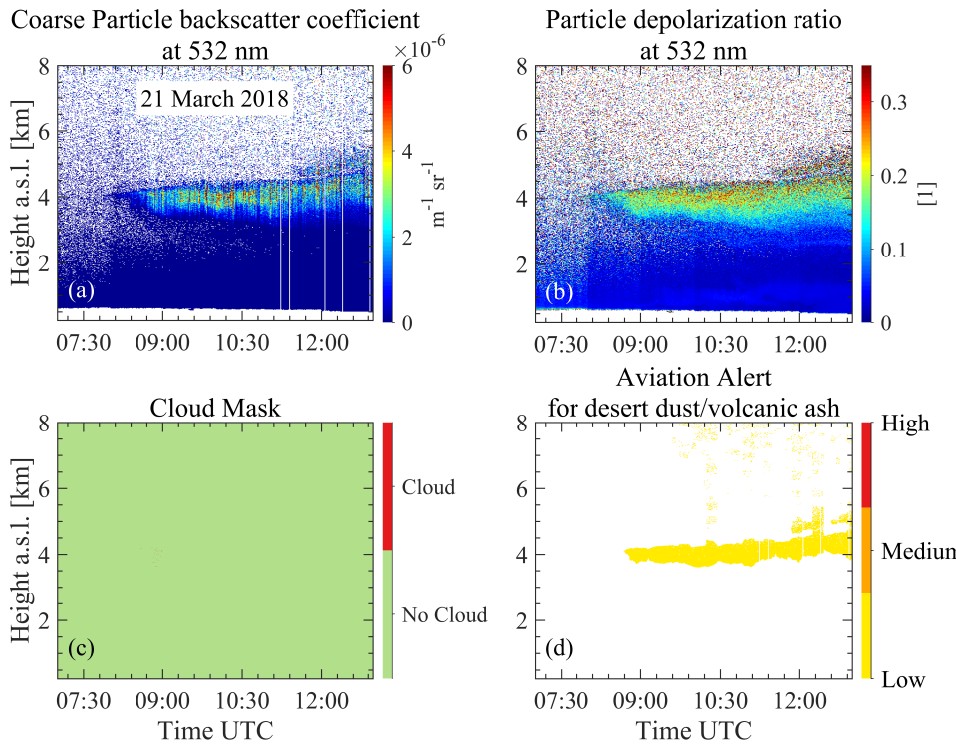

**Figure 5.** EARLINET observations at Finokalia on 21 March 2018: **(a)** the coarse particle backscatter coefficient at 532 nm, **(b)** the particle depolarization ratio at 532 nm, **(c)** the cloud screening output, and **(d)** the alert for aviation.

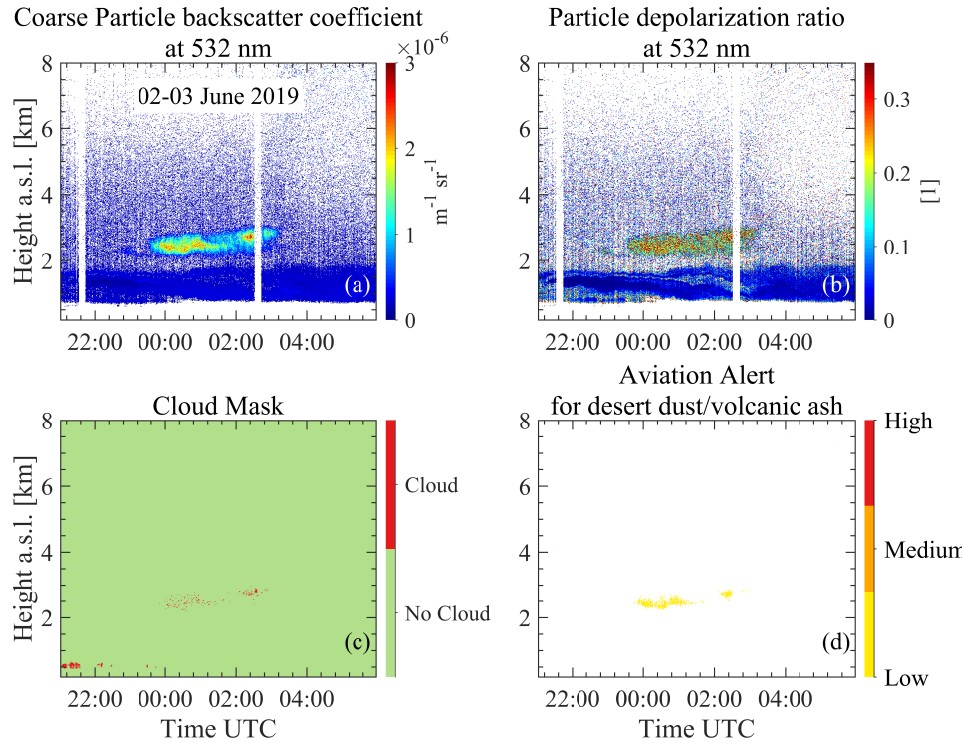

**Figure 6.** EARLINET observations at Antikythera on 2–3 June 2019. **(a)** the coarse particle backscatter coefficient at 532 nm, **(b)** the particle depolarization ratio at 532 nm, **(c)** the cloud screening output, and **(d)** the alert for aviation.



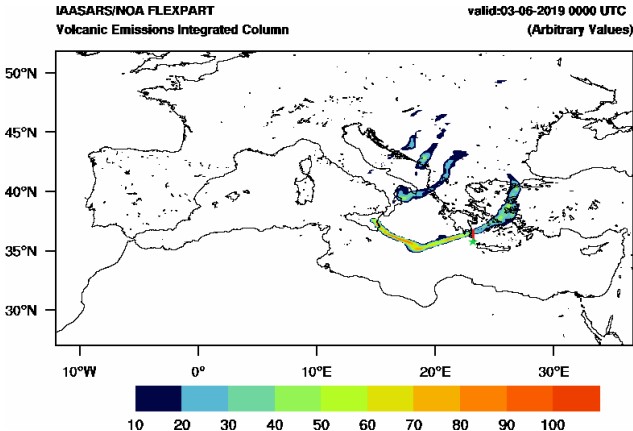

**Figure 7.** FLEXPART vertically integrated volcanic ash particles (arbitrary values) originating from Etna, 3 June 2019, 00:00 UTC. The green star indicates the location of Antikythera and the red line the misplacement of the simulated plume from the lidar station.





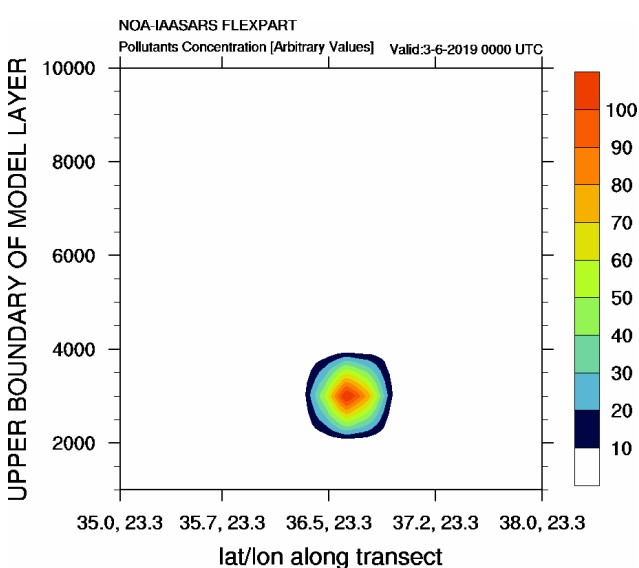

**Figure 8.** FLEXPART vertical cross-section of the simulated volcanic particles (in arbitrary values) over the greater Antikythera region. The exact location of the cross-section is indicated by the red line in Fig. 7.



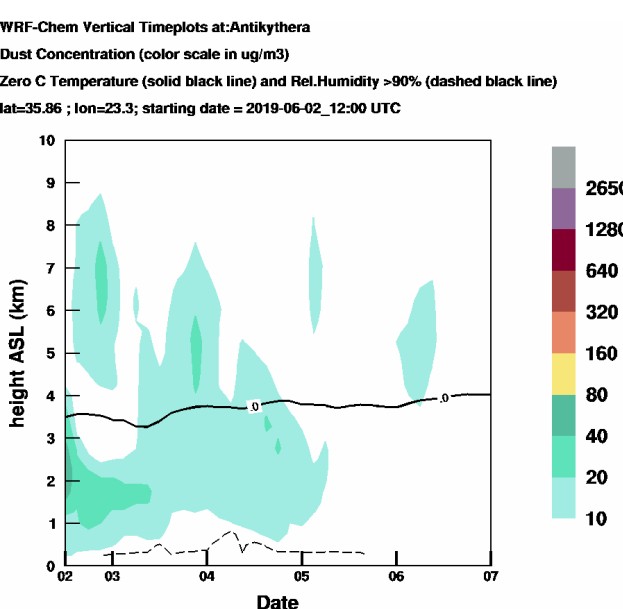

**Figure 9.** WRF-Chem time-height cross section of simulated dust concentration ($\mu g/m^3$) over Antikythera starting at 2 June 12:00 UTC. The solid black line is the 0 °C isotherm and the dashed black line indicates 90 % relative humidity.