# Peer review of "An EARLINET Early Warning System for atmospheric aerosol aviation hazards"

_Atmospheric Chemistry and Physics, 2020_

## Referee Comment (RC1) · Anonymous Referee #1 · 28 Mar 2020

[General comments]

This study well describes the development of early warning system using polarization lidars for aviation operations. Because elastic lidar data are used in the algorithm, the method can be applied not only for EARLINET lidars but also for aerosol lidars in the world. The method uses reasonable mass-to-extinction conversion factor and lidar ratios determined from plenty of references, and therefore the resulted threshold backscatters would be reliable. This study shows a good way of lidar data application and has a possibility to be a standard model of the warning system for decision making. I think this study satisfies the ACP publication level, but minor revisions should be needed because some points are still not clear as the following specific comments.

[Specific comments]

[Figure]

1. In Figure 5(d), there are mis-alerting points above 5 km. This would be caused by the low signal-to-noise ratio. The authors should discuss about this issue in the manuscript and please consider screening the mis-alerting points, for example, by using a threshold for signal-to-noise ratios. Signal averaging would also be helpful to decrease the false detection. Although the processed data can be available every hour (or possibly 30 minutes), the time resolution seems unnecessarily too high (I could not find the resolution used in your results). The authors should explain why such high resolution compared to the updating time (every hour) is needed without improving the signal-to-noise ratio by averaging.

2. In the second case of your results (observations at Antikythera), the authors mentioned that "few pixels within the same aerosol layer are wrongly classified as clouds". In Figure 6(d), are the cloudy pixels excluded from the aviation alert? Please clarify it because, if the cloudy pixels are not excluded, your system can easily misclassify cirrus clouds as dust.

3. In the last paragraph of the section 4.2, the authors mentioned that "In synthesis, both observations and model simulations advocate for the co-existence of volcanic dust and aged desert dust particles in the aerosol scene", but I could not understand this sentence, because, in the simulation results, volcanic ash dust did not appear below 2 km and desert dust particles are few above 2 km. Therefore, I supposed the volcanic ash dust and desert dust are not "co-existence" in the same layer. I believe these events happened at the same time, but the word "co-existence" may be misleading.

4. In Table 2, I understand that EWS was not available for the stations indicated by (*) because they could not provide depolarization channel during the exercise, but why it was not available for the other stations, for example Belgrade (SRB), even though the measurement performed percent was 100 %. The authors should mention the reason.

5. In Figure 9, the time domain is not same as Figure 6, so that comparison with the observation was not easy. Please consider changing the time domain or indicating

observation time domain by e.g., dashed lines.

[Technical corrections]

Page 3 line 23, "The latter and can be expressed as": "and" should be removed? Please confirm it.

---

## Referee Comment (RC2) · Anonymous Referee #2 · 15 Apr 2020

**Report on «An EARLINET Early Warming System for atmospheric aerosol aviation hazards», Papagiannopoulos et al.,**

**Anonymous referee**

**General comment:**

The paper authored by Papagiannopoulos et al. treats on methodology to detect airborne hazards for aviation in Near Real-Time (NRT). This methodology is mainly based on LiDAR network strategically deployed over the European region: EARLINET (European Aerosol LiDAR Network) network. The high resolution pre-processed data allows to obtain optical and microphysical parameters which could be helpful to develop the basis of a NRT Early Warning System (EWS) for aviation activity.

The aims of the paper are clearly written by the authors and focus on the natural hazards which impacted the aviation sector. As reported by the authors, the development of a NRT-EWS is a crucial point after the aviation crisis due the Eyjafjallajökull volcanic eruption in 2010. This paper is interesting about this point. The methodology described by the authors to develop a NRT-EWS from LiDAR observations are quite convincible. They evaluated this methodology by the analysis of two case studies. Nevertheless, the presentation of the results should be improved in order to help the reader to understand the conclusion of the study. The figures are fairly clear and helpful to support the key arguments provided in the paper. **I think that the manuscript may become acceptable after minor revisions.**

**Major Concerns:**

1) P8, line 10 to 18: The authors mentioned that the $C_v$ term can be estimated using AERONET observations. Given as the mass-to-extinction conversion factor is not a product provided by AERONET, it will be helpful to give some explanation and references on the methodology to obtain this parameter.

2) P9-10: It is clearly mentioned by the authors that the purpose of the paper is not to analyze in details these dust and volcanic events. Nevertheless, the transport analysis of the aerosols plumes should be improved. At least, it appears crucial to describe and show clearly the region impacted by the aerosols plume.

**Minors Concerns:**

1) P3, line 11: *"Nowadays, more than 30 stations are active and perform measurements according to the network's schedule (one daytime and two night-time measurements per week)"*. It could be interesting to include a map with localization of the sites involved in the network.

2) P3, line 22: *"To ensure homogeneous, traceable, and quality controlled analysis of raw lidar data across the network, a centralized and fully automated analysis tool, called the Single Calculus Chain (SCC), has been developed within EARLINET"*. Please, give references.

3) P10, line 10: *"Aerosol particles of possibly volcanic origin were monitored with the multi-wavelength lidar of NOA over Antikythera, Greece"*. Please, give references

4) P28: The quality of the figure 7 should be improved.

---

## Author Comment (AC1) · 2 Jun 2020

**Anonymous Referee 1:**

We would like to thank the referee for the careful reading and the insightful suggestions. RC is the referee comment and AR is the authors response. When needed, the part of the manuscript we modified or added to the old version is reported in bold.

**[Specific comments:]**

**RC:** 1. In Figure 5(d), there are mis-alerting points above 5 km. This would be caused by the low signal-to-noise ratio. The authors should discuss about this issue in the manuscript and please consider screening the mis-alerting points, for example, by using a threshold for signal-to-noise ratios. Signal averaging would also be helpful to decrease the false detection. Although the processed data can be available every hour (or possibly 30 minutes), the time resolution seems unnecessarily too high (I could not find the resolution used in your results). The authors should explain why such high resolution compared to the updating time (every hour) is needed without improving the signal-to-noise ratio by averaging.

**AR:** We fully agree with the referee that the temporal resolution is unnecessarily fine. Figures 5 and 6 are given in their full resolution – i.e., 7.5 m vertical resolution and 0.5 s temporal resolution, which apparently produced false alert pixels in the case of Finokalia (Figure 5d). The poor signal-to-noise ratio above the aerosol layer is creating this artefact. Therefore, we incorporated an averaging scheme for the lidar data. To this end, we applied the methodology with 5 min and 30 m averaged lidar data. Baars et al. (2017) used, for instance, 5 min temporal resolution and 30 m of vertical resolution for a similar application and system. The updated spatial resolution of the calibrated high-resolution data and the cloud screening output is explained in Page 5 directly below Lines 7-8:

**"The methodology to derive particle high-resolution data that is described in Sect. 3 is first cloud cleared and second is based on 5 min – 30 m averaged profiles in order to increase the signal-to-noise ratio."**

Furthermore, we implemented a spatial smoothing filter in the EWS product in order to further reduce spurious pixels that persist after the averaging. These random and isolated pixels are screened out by averaging the pixels in a small neighbourhood, typically a 3×3 dimension. The next few lines are added in the first paragraph of Page 9:

**"Furthermore, to avoid isolated false alarms in the EWS product we incorporated a linear spatial smoothing filter. It is the average of the pixels contained in the neighbourhood of each pixel, for which we defined a 3×3 pixel grid."**

Figures 1 and 2 show the improved Figures 5 and 6 of the manuscript. This averaging scheme and the new figures are inserted in the new version of the submitting paper.

[Figure]

*Figure 1: EARLINET observations at Finokalia on 21 March 2018: **(a)** the coarse particle backscatter coefficient at 532 nm, **(b)** the particle depolarization ratio at 532 nm, **(c)** the cloud screening output, and **(d)** the alert for aviation. Note that the cloud screening product is given in its full resolution – i.e., the vertical resolution is 7.5 m and the temporal resolution is 30 s – all the other products have resolution of 30 m and 5 min instead.*

[Figure]

*Figure 2: EARLINET observations at Antikythera on 2–3 June 2019. **(a)** the coarse particle backscatter coefficient at 532 nm, **(b)** the particle depolarization ratio at 532 nm, **(c)** the cloud screening output, and **(d)** the alert for aviation. Note that the cloud screening product is given in its full resolution – i.e., the vertical resolution is 7.5 m and the temporal resolution is 30 s – all the other products have resolution of 30 m and 5 min instead.*

**RC:** 2. In the second case of your results (observations at Antikythera), the authors mentioned that "few pixels within the same aerosol layer are wrongly classified as clouds". In Figure 6(d), are the cloudy pixels excluded from the aviation alert? Please clarify it because, if the cloudy pixels are not excluded, your system can easily misclassify cirrus clouds as dust.

**AR:** The referee is right as the whole methodology relies on the correct identification of cloudy pixels. Unfortunately, not clearly stated in the text, we kept the cloudy pixels in the aerosol layer that reside in the range 2-3 km, while the cloudy pixels in the lower part of the timeseries have been removed. In Section 2 we outlined that the cloud screening still suffers from false detection and for the cases shown in the submitted paper we manually examined the cloud mask and removed cloudy pixels when we considered necessary. The sentence in Page 10 Lines 18-19 is rephrased and now reads:

**"It is noteworthy that, as seen in the cloud mask, few pixels within the same aerosol layer are wrongly classified as clouds and are nonetheless used in the alert delivery".**

**RC:** In the last paragraph of the section 4.2, the authors mentioned that "In synthesis, both observations and model simulations advocate for the co-existence of volcanic dust and aged desert dust particles in the aerosol scene", but I could not understand this sentence, because, in the simulation results, volcanic ash dust did not appear below 2 km and desert dust particles are few above 2 km. Therefore, I supposed the volcanic ash dust and desert dust are not "co-existence" in the same layer. I believe these events happened at the same time, but the word "co-existence" may be misleading.

**AR:** We are sorry for the confusing statement. We wanted to stress the probable identification in the same aerosol scene of desert dust and volcanic dust in separate layers. Therefore, we modified the sentence as follows:

**"In synthesis, both observations and model simulations advocate for the identification of likely volcanic dust and aged desert dust particles in the same aerosol scene but in separate layers".**

**RC:** In Table 2, I understand that EWS was not available for the stations indicated by (*) because they could not provide depolarization channel during the exercise, but why it was not available for the other stations, for example Belgrade (SRB), even though the measurement performed percent was 100 %. The authors should mention the reason.

**AR:** We thank the referee for the opportunity to clarify that we were able to deliver the tailored product whenever possible. However, the EWS delivers no warning for aerosol free conditions and spherical particles layers – i.e., local pollution for most of the cases, therefore we chose not to show any results in the submitting manuscript. Moreover, we think that the examined cases (Antikythera and Finokalia) provide the concept of the methodology and showcase the performance in case of the above-mentioned conditions. The next sentence is inserted in Page 11 Line 13:

**"Hence, results of the exercise are not shown here, nonetheless the EARLINET observations are available through the EARLINET Quicklook Interface."**

Regarding the Belgrade station, there is a mistake as this EARLINET station has no depolarization channel. The table is updated, and the error corrected.

In the following, we show an example of the EWS during the exercise. Figure 3 illustrates the measurements from the Barcelona EARLINET site on 5 March from 14:00 UTC to 15:00 UTC, where a thin depolarizing layer resides slightly over the planetary boundary layer. The EWS produced no warning as the criteria were not met.

[Figure]

***Figure 3:*** *EARLINET observations at Barcelona on 5 March 2019. **(up left)** the coarse particle backscatter coefficient at 532 nm, **(up right)** the particle depolarization ratio at 532 nm, and **(down left)** the alert for aviation.*

**RC:** In Figure 9, the time domain is not same as Figure 6, so that comparison with the observation was not easy. Please consider changing the time domain or indicating observation time domain by e.g., dashed lines.

**AR:** Thank you. The figure now reports red lines that correspond to the time domain of the lidar observations. The figure is given below.

**[Technical corrections]**

**RC:** Page 3 line 23, "The latter and can be expressed as": "and" should be removed? Please confirm it.

**AR:** Thank you. This is a typo and it is corrected.

[Figure]

**WRF-Chem Vertical Timeplots at:Antikythera**

**Dust Concentration (color scale in ug/m3)**

**Zero C Temperature (solid black line) and Rel.Humidity >90% (dashed black line)**

**lat=35.86 ; lon=23.3; starting date = 2019-06-02_12:00 UTC**

***Figure 4:*** *WRF-Chem time-height cross section of simulated dust concentration (µg/m³) over Antikythera starting at 2 June 12:00 UTC. The solid black line is the 0 °C isotherm and the dashed black line indicates 90% relative humidity. The red lines correspond to time domain of the lidar observations – i.e., starting 21:00 UTC on 2 June 2019 until 06:00 UTC on 3 June 2019.*

---

## Author Comment (AC2) · 2 Jun 2020

**Anonymous Referee 2:**

We would like to thank the referee for the interesting and valuable comments and suggestions. RC is the referee comment and AR is the authors response. When needed, the part of the manuscript we modified or added to the old version is reported in bold. Moreover, the references of the cited literature are given in the end of the document.

**Major Concerns:**

**RC:** 1) P8, line 10 to 18: The authors mentioned that the Cv term can be estimated using AERONET observations. Given as the mass-to-extinction conversion factor is not a product provided by AERONET, it will be helpful to give some explanation and references on the methodology to obtain this parameter.

**AR:** The mass-to-extinction conversion factor can be obtained from the ratio of the coarse column volume concentration to the coarse mode aerosol optical thickness – i.e., $v_c/\tau_c$, both available on the AERONET database. The 532 nm AOT ($\tau_{c,532}$) is obtained from the coarse-mode 500 nm AOT by means of the respective Ångström exponent (AE). If a series of photometer measurements are available, the temporal average of this quantity over stable conditions can be used to get a better estimate of the needed ratio (Ansmann et al., 2012).

There are two different methods for calculating this ratio from AERONET data. Both methods make use of the aerosol size distribution to retrieve the volume of the coarse mode (Dubovik et al., 2006; Dubovik and King, 2000), and they differentiate in the way the coarse mode aerosol optical depth is calculated. The Spectral Deconvolution Algorithm (SDA) introduced by O'Neill et al. (2003) separates optically the fine and coarse mode while in the microphysical retrieval (Dubovik et al., 2006) a size threshold is invoked.

The conversion factor is central to the POLIPHON method and extended details of the AERONET data processing steps can be found in Mamouri and Ansmann (2014, 2015, 2016, 2017). Ansmann et al. (2019) further propose a methodology to derive climatological robust conversion factors. The data employed from the AERONET database are the coarse-mode volume concentration and the 500 nm AOT (denoted as extinction AOT in the database) together with the respective Ångström exponent for the 440-870 nm range. AOT and AE thresholds are used to screen out non-coarse particles and then the $\tau_{c,532}$ is estimated as described above. Finally, the climatological conversion factor is estimated.

The next few lines are inserted in the updated version of the submitting paper:

**"The term $c_v$ can be estimated using AERONET observations, being the ratio of the coarse column volume concentration ($v_c$) to the coarse mode aerosol optical thickness ($\tau_c$). More information on the different retrievals and AERONET data processing can be found in Ansmann et al. (2012), Mamouri and Ansmann (2017), and Ansmann et al. (2019)."**

**RC:** 2) P9-10: It is clearly mentioned by the authors that the purpose of the paper is not to analyze in details these dust and volcanic events. Nevertheless, the transport analysis of the aerosols plumes should be improved. At least, it appears crucial to describe and show clearly the region impacted by the aerosols plume.

**AR:** Indeed, the detailed analysis of transport processes is not the main purpose of the paper. However, we agree with the referee that more information would probably be useful specifically on the horizontal extent of the plumes.

To this direction, Hysplit (Stein et al., 2015) 2-day backward trajectories are superimposed on the Figure 4 of the submitting paper (i.e., the dust SEVIRI product) that illustrate the movement of the air-masses laden with Saharan dust particles towards the EARLINET station of Finokalia (Figure 1). Furthermore, we have included an additional modeling plot (Figure2) to better display the properties of the dust event. The corresponding few lines are also added in the Page 9 Line 23 of the submitting paper:

**"The extent of the dust layer at 12:00 UTC on 21 March 2018 is also evident at the WRF-CHEM dust optical depth (DOD) in Figure4b. The entire Eastern Mediterranean is affected by this episode and the simulated DOD exceeds 0.4 over certain parts of eastern Crete near the Finokalia station".**

[Figure]

**Figure 1:** The dust SEVIRI product (Marchese et al., 2017) at 12:00 UTC on 21 March 2018 is represented in confidence levels (i.e., brown pixels refer to high confidence and orange pixels to mid-low confidence). The grey pixels indicate the cloud cover. Additionally, the lines indicate the 2-day Hysplit backward trajectory analysis for airmasses arriving at 4 km a.s.l. over Finokalia on 21 March 2018. The symbols show the 6 h model output.

[Figure]

**Figure 2:** WRF-CHEM dust optical depth (DOD) on 21 March 2018 12:00 UTC.

Regarding the Etna dispersion case, the panel of Figure 3 shows the FLEXPART simulations of vertically integrated volcanic ash particles starting on 04:00 UTC of 30 May 2019. The output is given every 12 hours and illustrates the eastward movement of ash clouds since the eruption of Mt Etna in the early hours of 30 May. However, the detailed figure will not be included in the updated version of the submitting paper. Nonetheless, we added the following sentence in Page 10 - Line23 to better describe the event:

**"As shown by the FLEXPART simulation, this plume propagated eastwards from Sicily towards the Ionian Sea, reaching parts of South Greece".**

[Figure]

**Figure 3:** Volcanic ash particles simulated with FLEXPART originating from Etna, 30 May 2019, 004:00 UTC (output every 12 h).

**Minor Concerns**

**RC:** 1) P3, line 11: *"Nowadays, more than 30 stations are active and perform measurements according to the network's schedule (one daytime and two night-time measurements per week)".* It could be interesting to include a map with localization of the sites involved in the network.

**AR:** Figure 4 illustrates the network's geographic extent and the location of the active EARLINET stations (green squares) and the joining EARLINET stations (yellow squares) together with the non-active site of Finokalia (red square), for which lidar data are used in this study. In total, the map lists 36 active EARLINET stations, for more information see www.earlinet.org. The figure is inserted in the updated version of the submitting paper.

[Figure]

**Figure 4:** The EARLINET network. The green squares indicate the active stations, the yellow squares indicate the joining stations, and the red square indicates the non-active Finokalia (Greece) station.

**RC:** 2) P3, line 22: *"To ensure homogeneous, traceable, and quality controlled analysis of raw lidar data across the network, a centralized and fully automated analysis tool, called the Single Calculus Chain (SCC), has been developed within EARLINET".* Please, give references.

**AR:** The Single Calculus Chain is introduced in D'Amico et al. (2015) and is discussed in detail in the companion papers of D'Amico et al. (2016) and Mattis et al. (2016). The references are acknowledged and inserted in the updated version of the submitting manuscript:

**RC:** 3) P10, line 10: *"Aerosol particles of possibly volcanic origin were monitored with the multi-wavelength lidar of NOA over Antikythera, Greece".* Please, give references

**AR:** Tropospheric winds can advect volcanic particles, volcanic $SO_2$, and secondary sulfate particles from the erupting Mount Etna to the east, as it was demonstrated by Hughes et al. (2016) and Zerefos

et al. (2006). The next sentence is rephrased, and the above-mentioned references are inserted in the text:

**"The eastward advection of volcanic particles from Mount Etna presents a common pathway and has been previously investigated by means of active remote sensing (e.g., Hughes et al., 2016; Zerefos et al., 2006)."**

**RC:** 4) P28: The quality of the figure 7 should be improved.

**AR:** We agree with the referee. Figure 5 is an improved version of the Figure 7 of the submitting paper.

[Figure]

**Figure 5:** FLEXPART vertically integrated volcanic ash particles (arbitrary values) originating from Etna, 3 June 2019, 00:00 UTC. The green star indicates the location of Antikythera and the red line the misplacement of the simulated plume from the lidar station.

**REFERENCES**

Ansmann, A., Seifert, P., Tesche, M., and Wandinger, U.: Profiling of fine and coarse particle mass: case studies of Saharan dust and Eyjafjallajökull/Grimsvötn volcanic plumes, Atmospheric Chemistry and Physics, 12, 9399–9415, https://doi.org/10.5194/acp-12-9399-2012, 2012.

Ansmann, A., Mamouri, R.-E., Hofer, J., Baars, H., Althausen, D., and Abdullaev, S. F.: Dust mass, cloud condensation nuclei, and icenucleating particle profiling with polarization lidar: updated POLIPHON conversion factors from global AERONET analysis, Atmospheric Measurement Techniques, 12, 4849–4865, https://doi.org/10.5194/amt-12-4849-2019, 2019.

D'Amico, G., Amodeo, A., Baars, H., Binietoglou, I., Freudenthaler, V., Mattis, I., Wandinger, U., and Pappalardo, G.: EARLINET Single Calculus Chain–overview on methodology and strategy, Atmos. Meas. Tech., 8, 4891–4916, https://doi.org/10.5194/amt-8-4891-2015, 2015.

D'Amico, G., Amodeo, A., Mattis, I., Freudenthaler, V., and Pappalardo, G.: EARLINET Single Calculus Chain – technical – Part 1: Pre-processing of raw lidar data, Atmospheric Measurement Techniques, 9, 491–507, https://doi.org/10.5194/amt-9-491-2016, 2016.

Dubovik, O. and King, M. D.: A flexible inversion algorithm for retrieval of aerosol optical properties from Sun and sky radiance measurements, J. Geophys. Res., 105, 20673–20696, 2000.

Dubovik, O., Sinyuk, A., Lapyonok, T., Holben, B. N., Mishchenko, M., Yang, P., Eck, T. F., Volten, H., Muñoz, O., Veihelmann, B., van der Zande, W. J., Leon, J.-F., Sorokin, M., and Slutsker, I.: Application of spheroid models to account for aerosol particle nonsphericity in remote sensing of desert dust, J. Geophys. Res., 111, D11208, doi:10.1029/2005JD006619, 2006.

Hughes, E. J., Yorks, J., Krotkov, N. A., da Silva, A. M., and McGill, M.: Using CATS near-real-time lidar observations to monitor and constrain volcanic sulfur dioxide ($SO_2$) forecasts, *Geophys. Res. Lett.*, 43, 11,089-11,097, doi:10.1002/2016GL070119, 2016.

Mamouri, R. E. and Ansmann, A.: Fine and coarse dust separation with polarization lidar, Atmospheric Measurement Techniques, 7, 3717–3735, https://doi.org/10.5194/amt-7-3717-2014, 2014.

Mamouri, R. E. and Ansmann, A.: Estimated desert-dust ice nuclei profiles from polarization lidar: methodology and case studies, Atmos. Chem. Phys., 15, 3463–3477, https://doi.org/10.5194/acp-15-3463-2015, 2015.

Mamouri, R.-E. and Ansmann, A.: Potential of polarization lidar to provide profiles of CCN- and INP-relevant aerosol parameters, Atmos. Chem. Phys., 16, 5905–5931, https://doi.org/10.5194/acp-16-5905-2016, 2016.

Mamouri, R.-E. and Ansmann, A.: Potential of polarization/Raman lidar to separate fine dust, coarse dust, maritime, and anthropogenic aerosol profiles, Atmospheric Measurement Techniques, 10, 3403–3427, https://doi.org/10.5194/amt-10-3403-2017, 2017.

Marchese, F., Sannazzaro, F., Falconieri, A., Filizzola, C., Pergola, N., and Tramutoli, V.: An Enhanced Satellite–Based Algorithm for Detecting and Tracking Dust Outbreaks by Means of SEVIRI Data, Remote Sensing, 9, https://doi.org/10.3390/rs9060537, 2017.

Mattis, I., D'Amico, G., Baars, H., Amodeo, A., Madonna, F., and Iarlori, M.: EARLINET Single Calculus Chain – technical – Part 2: Calculation of optical products, Atmospheric Measurement Techniques, 9, 3009–3029, https://doi.org/10.5194/amt-9-3009-2016, 2016.

O'Neill, N. T., Eck, T. F., Smirnov, A., Holben, B. N., and Thulasiraman, S.: Spectral discrimination of coarse and fine mode optical depth, *J. Geophys. Res.*, 108, 4559, doi:10.1029/2002JD002975, D17, 2003.

Stein, A.F., Draxler, R.R, Rolph, G.D., Stunder, B.J.B., Cohen, M.D., and Ngan, F.: NOAA's HYSPLIT atmospheric transport and dispersion modeling system, Bull. Amer. Meteor. Soc., 96, 2059-2077, http://dx.doi.org/10.1175/BAMS-D-14-00110.1, 2015.

Zerefos, C., Nastos, P., Balis, D., Papayannis, A., Kelepertsis, A., Kannelopoulou, E., Nikolakis, D., Eleftheratos, C., Thomas, W., and Varotsos, C.: A complex study of Etna's volcanic plume from ground-based, *in situ* and space-borne observations, International Journal of Remote Sensing, 27:9, 1855-1864, DOI: 10.1080/01431160500462154, 2006.